# Characteristics of *Staphylococcus aureus* Isolated from Patients in Busia County Referral Hospital, Kenya

**DOI:** 10.3390/pathogens11121504

**Published:** 2022-12-09

**Authors:** Benear Apollo Obanda, Elizabeth A. J. Cook, Eric M. Fèvre, Lilly Bebora, William Ogara, Shu-Hua Wang, Wondwossen Gebreyes, Ronald Ngetich, Dolphine Wandede, Johnstone Muyodi, Beth Blane, Francesc Coll, Ewan M. Harrison, Sharon J. Peacock, George C. Gitao

**Affiliations:** 1Department of Veterinary Pathology, Microbiology and Parasitology, University of Nairobi, Nairobi P.O. Box 29053-00625, Kenya; 2Global One Health Initiative, Office of International Affairs, The Ohio State University, Columbus, OH 43210, USA; 3Centre for Microbiology Research Nairobi, Kenya Medical Research Institute, Nairobi P.O. Box 54840-00200, Kenya; 4International Livestock Research Institute, Nairobi P.O. Box 30709-00100, Kenya; 5Institute of Infection, Veterinary & Ecological Sciences, Leahurst Campus, University of Liverpool, Chester High Road, Neston CH64 7TE, UK; 6Department of Public Health, Pharmacology and Toxicology, University of Nairobi, Nairobi P.O. Box 29053-00625, Kenya; 7Division of Infectious Disease, Department of Internal Medicine, The Ohio State University, Columbus, OH 43210, USA; 8Department of Veterinary Preventive Medicine, College of Veterinary Medicine, The Ohio State University, Columbus, OH 43210, USA; 9The Centre for Infectious and Parasitic Diseases Control Research, Busia P.O. Box 3-50400, Kenya; 10Department of Medicine, University of Cambridge, Cambridge CB2 2QQ, UK; 11Department of Infection Biology, London School of Hygiene and Tropical Medicine, University of London, London WC1E 7HT, UK; 12Wellcome Sanger Institute, Hinxton CB10 1SA, UK; 13Department of Public Health and Primary Care, University of Cambridge, Cambridge CB1 8RN, UK

**Keywords:** *S. aureus*, MSSA, MRSA, hospital, Kenya, antimicrobial resistance

## Abstract

*Staphylococcus aureus* is an important pathogen associated with hospital, community, and livestock-acquired infections, with the ability to develop resistance to antibiotics. Nasal carriage by hospital inpatients is a risk for opportunistic infections. Antibiotic susceptibility patterns, virulence genes and genetic population structure of *S. aureus* nasal isolates, from inpatients at Busia County Referral Hospital (BCRH) were analyzed. A total of 263 inpatients were randomly sampled, from May to July 2015. The majority of inpatients (85.9%) were treated empirically with antimicrobials, including ceftriaxone (65.8%) and metronidazole (49.8%). Thirty *S. aureus* isolates were cultured from 29 inpatients with a prevalence of 11% (10.3% methicillin-susceptible *S. aureus* (MSSA), 0.8% methicillin resistant *S. aureus* (MRSA)). Phenotypic and genotypic resistance was highest to penicillin-G (96.8%), trimethoprim (73.3%), and tetracycline (13.3%) with 20% of isolates classified as multidrug resistant. Virulence genes, Panton-Valentine leukocidin (*pvl*), toxic shock syndrome toxin-1 (*tsst-1*), and *sasX* gene were detected in 16.7%, 23.3% and 3.3% of isolates. Phylogenetic analysis showed 4 predominant clonal complexes CC152, CC8, CC80, and CC508. This study has identified that inpatients of BCRH were carriers of *S. aureus* harbouring virulence genes and resistance to a range of antibiotics. This may indicate a public health risk to other patients and the community.

## 1. Introduction

*Staphylococcus aureus* can asymptomatically colonize healthy individuals, and carriers are at higher risk of a variety of serious infections [1]. Nasal carriage has been associated with nosocomial infections with potential to cause severe morbidity and mortality [2,3]. The pathogen is a common cause of skin and soft tissue infections in humans and animals; it can cause food poisoning and more serious conditions such as pneumonia, endocarditis, osteomyelitis, sepsis, and toxic shock syndrome [4].

*S. aureus* has a wide range of virulence factors including toxic shock syndrome toxin-1 (TSST-1) encoded by the *tsst-1* gene, which is associated with most cases of menstrual Toxic Shock Syndrome and half of non-menstrual Toxic Shock Syndrome cases [5]. A virulent cytotoxic molecule, Panton-Valentine leukocidin encoded by the *pvl* gene, can cause destruction of leukocytes and tissue necrosis and is often associated with skin and soft tissue infections [6,7]. Some methicillin-resistant *S. aureus* (MRSA) strains have been shown to harbor a virulence factor *sasX*, hosted on a mobile genetic element. This factor has been demonstrated in vitro and in vivo (mice) to contribute to colonization and pathogenesis, by significantly promoting nasal colonization, lung disease and abscess formation; it also supports immune system evasion [8]. Virulence factors associated with human diseases can be detected by identifying the encoding genes using molecular methods; however, there is minimal data available on virulence factors present in *S. aureus* from Kenya.

*S. aureus* is notorious for its ability to develop resistance to antibiotics, originally to penicillin and methicillin and most recently to linezolid, vancomycin, teicoplanin, and daptomycin [9], with MRSA becoming more common. Some strains have become resistant to more than one class of antibiotics, converting these strains to being multidrug-resistant (MDR) [10]. The emergence of antibiotic resistance, coupled with the shortage of newly developed antibiotics, is a major threat to the health of both humans and animals [11]. Excessive use of antimicrobial empirical therapy in hospital settings has been reported as a contributor to the emergence of bacterial resistance to antimicrobials [12,13]. Antimicrobial-resistant *S. aureus* infections have reached epidemic levels [14,15], necessitating the study of its global epidemiology [16,17,18].

Clonal complex 5 (CC5) (Pediatric NY Japan- ST5) and CC8 (Iberian Hanover-ST 8, 239, 247, and 250) are the most prevalent lineages of *S. aureus* globally; others include CC22 (EMRSA-15 Barnum-ST 22), CC30 (EMRSA-16 USA200-ST 36, 30) and CC45 (USA600 Berlin-ST 45) [19,20,21]. In addition, in the last two decades, there have been reports on the spread of livestock-associated clonal complex CC398; this is an emerging problem in many parts of the world [22]. Methicillin-susceptible *S. aureus* (MSSA) isolated in Africa mainly belong to CC5, CC152, CC30, and CC15, whereas CC8 (ST239 subgroup), CC5 (ST5), and CC30 (ST36) lineages are highly prevalent in Central, West and South Africa [17,23].

The purpose of this study was (1) to assess empirical antibiotic therapy use in patients at Busia County Referral Hospital (BCRH), (2) determine the prevalence of *S. aureus* isolated from the patients’ nasal swabs and respective antibiotic susceptibility patterns, (3) demonstrate presence of resistance and virulence genes, and (4) identify the *S. aureus* clones present using a whole-genome sequencing approach.

## 2. Materials and Methods

### 2.1. Study Site

The study site was BCRH, the largest health facility in Busia County. Busia County has a total of 81 health facilities, including BCRH, 6 sub-county hospitals, 12 health centers and 49 dispensaries. BCRH has a bed capacity of 160 across 7 wards: pediatrics, male medical, female medical, female surgical and gynecology, male surgical, maternity and a newborn unit. The hospital admits 40–50 patients per day and the average stay is 4 days [24].

### 2.2. Study Population

The study was conducted between May and July 2015. A total of 263 inpatients participated in the study; this constituted 10% of the 2674 inpatients in BCRH during the study period. This study was based on a cross-sectional random sampling design. Admitted patients who were eighteen years of age and above and able to sign the consent form were included. An assent form for children less than 18 years was signed by a parent or legal guardian. Inpatients must have been admitted for 48 h or longer, and inpatients who had recent nasal surgery or other medical conditions were excluded from the study.

### 2.3. Sample Collection, Handling, Bacterial Isolation and Identification

All participants were informed of the project objectives and protocol by medical and clinical officers who collected signed informed consent. Nasal samples were collected by rotating a sterile swab five times in both anterior nares, from consenting inpatients. The swabs were inoculated in tryptone soya broth/6% salt and transported in cool boxes to the lab for culturing.

The swabs were streaked on mannitol salt agar (MSA) and incubated at 37 °C overnight. Suspect *S. aureus* colonies (those that fermented mannitol, producing yellow colonies) were stocked in tryptone soya broth with 10% glycerol and stored at −40 °C; they were later transported on dry ice to Kenya Medical Research Institute (KEMRI) laboratories in Nairobi. The presumptive *S. aureus* isolates were further cultured onto MSA and repeatedly sub-cultured to get pure culture. The *S. aureus* isolates were identified using Gram reaction (Gram-positive cocci in clumps), catalase, coagulase (tube method using rabbit plasma) and DNase tests.

When stocking the pure *S. aureus* growth, a long sweep of the colonies was done to allow preservation of genetic diversity of nasal carriage of the participant. During whole genome sequencing of the sample, if sequences from multiple isolates were detected, these samples were recultured and single colonies selected for sequencing.

### 2.4. Antimicrobial Susceptibility Testing

Antimicrobial susceptibility was also performed using the VITEK 2 instrument (bioMerieux, Marcy-l’Étoile, France), for benzylpenicillin, cefoxitin, oxacillin, ciprofloxacin, erythromycin, chloramphenicol, daptomycin, fusidic acid, gentamicin, linezolid, mupirocin, nitrofurantoin, rifampicin, teicoplanin, tetracycline, tigecycline, trimethoprim, vancomycin, clindamycin, and inducible resistance to clindamycin. Multi-drug resistant *S. aureus* were defined as isolates that were resistant to three or more antimicrobials [25,26,27].

### 2.5. Molecular Genotype Testing

DNA extraction from *S. aureus* isolates was performed using QIAGEN DNeasy^®^ Blood & Tissue Kit (QIAGEN, Valencia, CA, USA) at KEMRI, Nairobi. Staphylococcal cassette chromosome (SCC) mec typing was performed using previously described methods [28]. Isolates were screened for *pvl* and *tsst-1* genes by PCR using previously described oligonucleotide primers [29,30].

DNA extraction from *S. aureus* isolates was performed on a QIAcube, using the QIAamp 96 HT kit (QIAGEN). Genomic libraries were generated and sequenced on an Illumina HiSeq 2000 (Illumina Inc., San Diego, CA, USA) at the Wellcome Sanger Institute, Hinxton, UK. Illumina reads were analysed based on the *S. aureus* MLST database (https://pubmlst.org/organisms/staphylococcus-aureus, accessed on 31 March 2021) [31], analysis of virulence and antimicrobial resistance genes were conducted using the virulence finder database (https://cge.cbs.dtu.dk/services/, accessed on 31 March 2021).

### 2.6. Genomic Analyses

Paired-end Illumina reads were mapped to the *S. aureus* reference genome ST22 strain HO 5096 0412 (accession number HE681097) using Snippy v4.6.0 (https://github.com/tseemann/snippy, accessed on 31 October 2022). Whole-genome alignments were created by keeping a version of the reference genome with only substitution variants replaced (i.e., SNPs but not indels) using Snippy’s.consensus.subs.fa output files. The *S. aureus* species core-genome had been previously derived [32] from a collection of 800 *S. aureus* from multiple host species [33]. The portion of the reference genome (2.83 Mb) corresponding to the core genome (1.76 Mb) was kept from whole-genome alignments and used to generate maximum likelihood trees using IQ-TREE v1.6.10 with default settings. The resulting core-genome phylogeny was plotted with isolate metadata using ggtree v.3.0.4 [34] and ggtreeExtra v.1.2.3 on R v4.1.0 [35].

### 2.7. Ethical Approval

The study was approved by the Centre for Microbiology Research Centre Scientific Committee, Kenya Medical Research Institute scientific steering Committee and Ethical Review Committee (SSC No 2944, granted 13 May 2015).

## 3. Results

### 3.1. Patient Data

Sampling was conducted between 21 May and 7 July 2015. A total of 263 patients were recruited into the study, 141 females (53.6%) and 122 males (46.4%). The majority of patients were in the surgical ward (119/263, 45.2%), followed by the medical ward (91/263, 34.6%) and the pediatric ward (52/263, 19.8%) and 1 patient on the private ward. The average (mean) time spent in hospital was 8.8 days with a median stay of 5 days, with the longest stay being 103 days. The longest average inpatient stays were on the surgical ward 12.3 days, while average inpatient stays on the medical ward were 5.9 days and pediatric ward 5.8 days.

### 3.2. Empirical Antibiotic Therapy

A total of 226/263 (85.9%) inpatients were treated empirically using various antimicrobials. Most patients (150/263, 57.0%) received 2 antimicrobials in combination. Ceftriaxone, a third-generation cephalosporin, was the most prescribed antimicrobial in 173/263 (65.8%) inpatients; it was prescribed to 21/53 (39.6%) pediatric, 56/91 (61.5%) medical, and 96/119 (80.7%) surgical patients (Table 1). Metronidazole was the second-most prescribed antimicrobial to 131/263 (49.8%) of inpatients; 14/53 (26.4%) pediatric, 56/91 (24.2%) medical, and 95/119 (79.8%) surgical patients. The combination of ceftriaxone and metronidazole was prescribed to 111/263 (42.2%) patients; predominantly to patients in the surgical ward 84/119 (70.6%). The third-most-prescribed antimicrobial was penicillin-G (21/263, 8%); predominantly to pediatric patients (17/53, 32.1%). The next commonly used antibiotic in pediatric patients was gentamicin (10/53, 18.9%). Ciprofloxacin and erythromycin were less commonly used at 12/263 (4.6%) and 7/263 (2.7%), respectively and predominantly given to medical patients. Four patients were receiving combination therapy rifampin, isoniazid, pyrazinamide, and ethambutol (RHZE) for tuberculosis. Other antimicrobials used less frequently included: sulfamethoxazole/trimethoprim (*n* = 6), amoxicillin (*n* = 3), tinidazole (*n* = 3), clindamycin (*n* = 3), ampicillin (*n* = 2), doxycycline (*n* = 1), clarithromycin (*n* = 1), norfloxacin (*n* = 1) and flucloxacillin (*n* = 1).

### 3.3. Staphylococcus aureus (MRSA and MSSA) Isolated from Hospital Patients

Nasal swab samples were collected from 263 patients at BCRH; samples were obtained from inpatients who had been admitted for 48 h or longer. Of these 29 (11.0%) were colonized with *S. aureus*. One patient was colonized with two different *S. aureus* sequence types. Male patients (19/122, 15.6%) were more frequently colonized than female patients (10/141, 7.1%). Colonization rates of patients on the different wards were: pediatrics 8/53 (15.1%), medical 7/91 (7.7%) and surgical 14/119 (11.8%).

### 3.4. Antimicrobial Resistance Profiles of the Isolated Staphylococcus aureus (MRSA and MSSA)

Antimicrobial susceptibility testing was conducted on 30 *S. aureus* isolates from 29 patients, of which 2 (6.7%) were MRSA. Nearly all *S. aureus* strains (29/30, 96.7%) were phenotypically resistant to penicillin-G, 22/30 (73.3%) were resistant to trimethoprim, and 4/30 (13.3%) were resistant to tetracycline. There was phenotypic resistance to ciprofloxacin (2/30, 6.7%), clindamycin (6.7%), vancomycin (6.7%), and erythromycin (6.7%), cefoxitin (6.7%), oxacillin (6.7%), and gentamicin (1/30, 3.3%) (Figure 1). All isolates were susceptible to chloramphenicol, daptomycin, fusidic acid, linezolid, mupirocin, nitrofurantoin, rifampicin, teicoplanin and tigecycline.

Twenty percent (6/30) of *S. aureus* isolates were phenotypically multidrug-resistant, the majority of which (4/6, 66.7%) were resistant to penicillin-G-trimethoprim-tetracycline (Figure 1). Highly resistant isolates (resistant to more than 5 classes of antibiotics) were also detected, including two MRSA isolates from two inpatients and one MSSA from one inpatient (Figure 1).

Genotypic resistance was detected using specific genes. The proportion of isolates with resistance to beta-lactamase (*blaZ*) was 30/30 (100%), tetracycline (*tetK*) 3/30 (10%), tetracycline (*tetM*) 2/30 (6.7%), trimethoprim (*dfrG*) 22/30 (73.3%), erythromycin (*ermA*) 1/30 (3.3%), erythromycin (*ermC*) 1/30 (3.3%), gentamicin (*aacA-aphD*) 1/30 (3.3%), and streptomycin (*aad9*) 1/30 (3.3%). The phenotypic and genotypic resistance patterns were almost 100% in agreement for all isolates.

There was no relationship between carriage of multi-drug resistant *S. aureus* and antimicrobial use. Of the 27 *S. aureus* isolates that were isolated from inpatients who had recently received antimicrobials 5 were MDR (18.5%) and of the three *S. aureus* isolates from inpatients who did not receive antimicrobials one was MDR (33.3%).

### 3.5. Clonal Complexes of the Staphylococcus aureus Isolates

Analysis of the 30 *S. aureus* isolates yielded 9 clonal complexes and 13 STs (Table 2). The predominant clonal complexes were CC152, CC8, CC80, and CC508. The predominant STs were ST 152 (*n* = 5), ST 8 (*n* = 5) and ST508 (*n* = 5) (Table 2). Other MSSA sequence types detected were ST5 (*n* = 1), ST22 (*n* = 2), ST25 (*n* = 1), ST80 (*n* = 3), ST188 (*n* = 2), ST573 (*n* = 1), ST580 (*n* = 2), ST1633 (*n* = 1). The MRSA sequence types were ST140 and a new sequence type ST241.

### 3.6. Virulence Factors in the Isolated Staphylococcus aureus

Six of 30 isolates were positive for the *pvl* gene (20%), associated with 3 sequence types (STs). The *pvl*-positive isolates belonged to ST152 (*n* = 4), ST5 (*n* = 1), and ST1633 (*n* = 1) (Table 2). Another 7 isolates tested positive for the *tsst-1* gene (23.3%); these belonged to ST508 (*n* = 5), and ST22 (*n* = 2). *SasX* virulence gene was detected in a single MRSA ST241.

### 3.7. MRSA Isolates

One strain was MRSA ST 241, a single-locus variant of ST239 carrying associated SCCmec type III and previously associated with hospital acquired infection (33). This was isolated from a female patient who had stayed in the hospital for 7 days, the variant exhibited multiple phenotypic drug resistance to penicillin, cefoxitin, oxacillin, ciprofloxacin, erythromycin, gentamicin, tetracycline and trimethoprim, and genotypic resistance to beta-lactamase (*blaZ*), cefoxitin (*mecA*), tetracyclin (*tetK*/*tetM*), trimethoprim (*dfrG*), erythromycin (*ermA*), gentamicin (*aacA-aphD*) and streptomycin/spectinomycin (*aad9*). This strain also had a *sasX* gene.

The other strain was MRSA ST140 strain harboring SCCmec type IV cassette with phenotypic resistance to penicillin, cefoxitin, oxacillin, tetracycline and trimethoprim and hosting resistance genes to beta-lactamase (*blaZ*), cefoxitin (*mecA*), tetracycline (*tetM*), and trimethoprim (*dfrG*). This was isolated an adult male patient who had stayed in the surgical ward for 6 days, the strain has previously been associated with hospital transmission (34).

## 4. Discussion

In this study, 263 inpatients were screened for *S. aureus* nasal carriage; the overall carriage prevalence was 11.0%. The nasal carriage rate was slightly higher than that reported by a similar study conducted in a mid-sized government hospital in peri-urban Kenya, where the nasal carriage rate of *S. aureus* was 8.9% [36]. The prevalence reported in this study is lower than studies in other regions of Africa: 14% in Accra, Ghana, 37% in Bobo Dioulasso, Burkina Faso, and 29% in Uganda [37,38,39]. Other regions with higher nasal carriage rates include the Netherlands (26%), and United States (24%) [40,41]. The reduced prevalence of *S. aureus* carriage may be due to the sampling and isolation methods used. Previous research has indicated that the type of swab and the culture media (mannitol salt agar) may reduce the isolation of *S. aureus* [42,43].

Empirical therapy was found to be administered to the majority of inpatients; the most commonly used antimicrobials being ceftriaxone and metronidazole. The pattern of antimicrobial use is consistent with previous research in hospitals in Kenya [44,45], and the proportion of patients receiving antibiotics in this study (85.9%) was similar to a report of 80% in a study by Maina et al., 2020 [46]. Empirical antibiotic treatment has been linked to increased prevalence of infections with MDR bacterial pathogens. These pathogens pose a considerable threat to global public health, especially in hospitals like BCRH, where they could be the driving force behind increased antimicrobial resistance towards commonly prescribed empirical antimicrobial therapy. This limits treatment options, especially when considering the slow rate of new antibiotic discoveries [47].

This study identified *S. aureus* isolates carrying genes encoding for 3 virulence factors (*pvl*, *tsst-1* and *sasX*). Busia County might be considered a *pvl*-endemic region since the prevalence estimate (20%) is consistent with *pvl*-positivity rates ranging from 17% to 74% reported in other parts of Africa [48,49,50]. Another virulence factor gene, *tsst-1*, was detected in almost one quarter of the *S. aureus* isolates, which is consistent with reports that *tsst-1* is present in between 5 to 25% of *S. aureus* strains [51]. The report of an MRSA isolate carrying the *sasX* virulence gene, in this study, is the first in Africa; indicating the high chances that this strain is present in other nosocomial MRSA strains in Africa. The *sasX* virulence gene plays an important role in the organism’s colonization and pathogenesis, by significantly promoting nasal colonization, lung disease and abscess formation; it also supports immune system evasion [8].

This study has revealed that the majority of *S. aureus* strains were susceptible to linezolid, clindamycin, gentamicin, ciprofloxacin, and erythromycin; similar results have been reported in other studies in Kenya [52,53]. Nearly all *S. aureus* strains (96.7%) were resistant to penicillin G, 73.3% were resistant to trimethoprim and 13.3% were resistant to tetracycline. Previous studies in Kenya have reported high *S. aureus* antimicrobial resistance to penicillin, trimethoprim and tetracycline [36,54,55]. The proportion of *S. aureus* isolates that were identified as MRSA (6.7%) was consistent with other studies in Kenya 7.0% [36] and elsewhere: 9.5% in Ghana [38] and 9.7% in Uganda [37]. We did not detect an association between antimicrobial use and MDR *S. aureus* in this study which may be due to the small number of samples. Future research may focus on the source and evolution of AMR bacteria in this ecosystem.

Two MDR MRSA strains were identified and a MDR MSSA strain. The presence of antibiotic resistant pathogenic bacteria hosting resistant genes may promote transmission of genes to other pathogens, leading to more resistant pathogens and emergence of multi drug resistant strains [56,57]. The widespread use of empirical antimicrobial therapy in BCRH is likely to drive the selection of the small fraction of MDR resistant strains that exist in the hospital setting microbiota, by exerting selective pressure on susceptible microorganisms, thus giving a survival advantage to resistant strains. This compounds the problem of antimicrobial resistance in this region [58] and may render antibiotics ineffective, narrowing the therapeutic value of available antibiotics in the hospital [59].

The two MDR MRSA strains were ST241 and ST140. ST241 has previously been detected in a Nigerian hospital and ST140 in a health care institution in Angola [60], inferring a possibility of international transmission of these strains into Busia County or vice versa.

The population structure of MSSA has been reported to be more diverse than that of MRSA globally [19]. High clonal diversity among MSSA isolates was observed in this study. Methicillin-sensitive *S. aureus* clonal complex 152 (ST152 and ST1633) was the predominant lineage; with the majority of them harboring *pvl* gene (83%). This clonal complex has been reported to be endemic in Africa and the Caribbean, as opposed to Europe [61,62]. Other commonly identified clonal complexes in Busia were CC8, CC80, and CC508. A recent review indicated that reported clonal complexes in Kenya are CC5, CC7, CC8, CC22, CC88 and CC152 [63]. Previous research by these authors demonstrated *S. aureus* isolates from abattoir workers in western Kenya were predominantly CC152 and CC8 [64].

At the time of this research there was no information regarding the circulating *S. aureus* strains in animals in Kenya. The role of animals in the epidemiology of MSSA and MRSA has been highlighted in other regions with the documentation of MRSA and MDR MSSA in animals [65,66], and transmission to people both with and without animal contact [67,68]. The unregulated use of antimicrobials in animal and human medicine may result in increased drug resistance posing a threat to public health. Increased surveillance is required to understand the epidemiology of AMR bacteria in this region to develop appropriate integrated control strategies.

## 5. Conclusions

This study has identified inpatients as important reservoirs of *S. aureus* organisms which are resistant to a range of antimicrobials, including MDR *S. aureus*; MRSA included. Some of these strains also harbor disease causing virulence factors. Nasal carriage of MRSA and MSSA is a potential reservoir of nosocomial infections for susceptible patients in hospitals, with a potential to cause severe morbidity and mortality. Data from the genetic structure of *S. aureus*, from the inpatients of Busia County contributes to knowledge on molecular epidemiology of *S. aureus* in this region and may be useful when developing prevention and containment measures.

## Figures and Tables

**Figure 1 pathogens-11-01504-f001:**
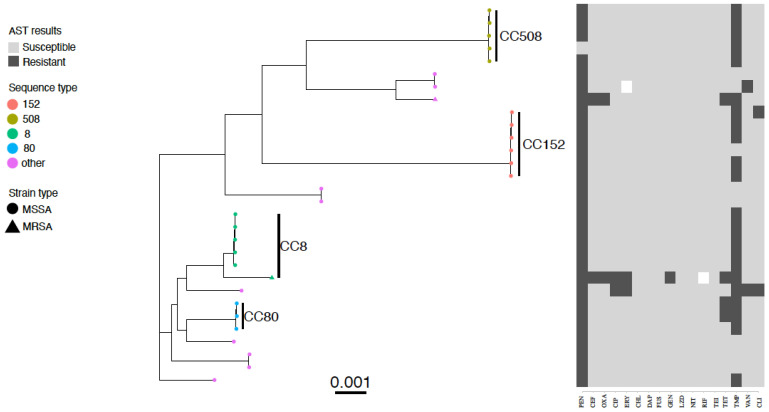
Core genome phylogenetic tree and antibiogram of MSSA and MRSA isolates colonizing inpatients at Busia County Referral Hospital, Kenya May–July 2015. Predominant ST types are differentiated by colour, MRSA isolates are indicated by a triangle symbol and MSSA by circles. Phenotypic antimicrobial resistance is indicated by dark grey bars (resistant), white bars (intermediate), and light grey (susceptible). Antimicrobials: PEN—benzylpenicillin, CEF—cefoxitin, OXA—oxacillin, CIP—ciprofloxacin, ERY—erythromycin, CHL—chloramphenicol, DAP—daptomycin, FUS—fusidic acid, GEN—gentamicin, LZD- linezolid, NIT—nitrofurantoin, RIF—rifampicin, TEI—teicoplanin, TET—tetracycline, TMP—trimethoprim, VAN—vancomycin, CLI—clindamycin.

**Table 1 pathogens-11-01504-t001:** Antimicrobials prescribed for inpatients in medical, surgical, and pediatric wards at Busia County Referral Hospital, May–July 2015.

	Medical *n* = 91	Surgical *n* = 119	Pediatrics and Amenity *n* = 53	Total *n* = 263
Treated with antibiotics	74 (81.3%)	110 (92.4%)	42 (79.2%)	226 (85.9%)
Treated with 2 antibiotics	35 (38.5%)	90 (75.6%)	25 (47.2%)	150 (57.0%)
Ceftriaxone	56 (61.5%)	96 (80.7%)	21 (39.6%)	173 (65.8%)
Metronidazole	22 (24.2%)	95 (79.8%)	14 (26.4%)	131 (49.8%)
Penicillin-G	2 (2.2%)	2 (1.7%)	17 (32.1%)	21 (8.0%)
Ciprofloxacin	10 (11.0%)	1 (0.8%)	1 (1.9%)	12 (4.6%)
Gentamicin			10 (18.9%)	10 (3.8%)
Erythromycin	7 (7.7%)			7 (2.7%)

**Table 2 pathogens-11-01504-t002:** Sequence types, Clonal complexes, and presence of Panton Valentine leukocidin (*pvl*) and Toxic shock syndrome toxin {*tsst-1*} genes in *Staphylococcus aureus* strains recovered from nasal samples of patients admitted at Busia County Referral Hospital, Kenya, May–July 2015 (*n* = 30). * Virulence-associated *sasX* gene.

Sequence Type	Clonal Complex	Total	PVL Positive	TSST Positive
ST152	CC152	5	4	
ST1633	CC152	1	1	
ST8	CC8	5		
ST80	CC80	3		
ST188	CC1	2		
ST573	CC1	1		
ST22	CC22	2		2
ST25	CC25	1		
ST508	CC508	5		5
ST580		2		
ST5	CC5	1	1	
ST241 *	CC8	1		
ST140	CC398	1		

## Data Availability

Data is contained within the article.

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
