# Peer review of "Characteristics of Staphylococcus aureus Isolated from Patients in Busia County Referral Hospital, Kenya"

_pathogens, 2022, doi:10.3390/pathogens11121504_

Round 1

Reviewer 1 Report

Obanda et al characterized Staphylococcus aureus isolated from patients in Busia County Referral Hospital, Kenya. Authors identified that in-patients of Busia hospital were carriers of S. aureus harboring virulence genes and resistance to a range of antibiotics. The manuscript is generally good and well-written. I have some comments:

i)                 Please expand the discussion to shed light on the role of animals in the transmission of such zoonotic pathogens due to the overuse or misuse of antibiotics in animal farms. Unfortunately, MDR  zoonotic bacteria were isolated and characterized from animals during the last few decades. Implementation of One Health and investigation of the epidemiology of these pathogens in patients are also very important. for examples. Benrabia  et al. Methicillin-Resistant Staphylococcus Aureus (MRSA) in Poultry Species in Algeria: Long-Term Study on Prevalence and Antimicrobial Resistance. Vet Sci. 2020 Apr 27;7(2):54.

Kalee et al. 2021. Prevalence and antimicrobial susceptibility profiles of Staphylococcus aureus from raw bovine milk in dairy and pastoral farms in Morogoro region, Tanzania. Ger.J. Vet. Res. (2): 1-7.

ii)                Tables and figures should be described to be stand-alone (figure/table legends)

iii)              Staphylococcus aureus should be italic throughout the manuscripts (See the attached file)

iv)              Also all genes should be in italics (See the attached file).

v)                All references should be revised according to the journal style.

Other minor comments, please see the attached file

Author Response

Many thanks for comments on our manuscript. We have responded to your comments below and in the manuscript.

i)                Please expand the discussion to shed light on the role of animals in the transmission of such zoonotic pathogens due to the overuse or misuse of antibiotics in animal farms. Unfortunately, MDR  zoonotic bacteria were isolated and characterized from animals during the last few decades. Implementation of One Health and investigation of the epidemiology of these pathogens in patients are also very important. for examples.

Benrabia  et al. Methicillin-Resistant Staphylococcus Aureus (MRSA) in Poultry Species in Algeria: Long-Term Study on Prevalence and Antimicrobial Resistance. Vet Sci. 2020 Apr 27;7(2):54.

Kalee et al. 2021. Prevalence and antimicrobial susceptibility profiles of Staphylococcus aureus from raw bovine milk in dairy and pastoral farms in Morogoro region, Tanzania. Ger.J. Vet. Res. (2): 1-7.

 Many thanks for this suggestions and we have added these details and references to the discussion.

“At the time of this research there was no information regarding the circulating S. aureus strains in animals in Kenya. The role of animals in the epidemiology of MSSA and MRSA has been highlighted in other regions with the documentation of MRSA and MDR MSSA in animals [65,66], and transmission to people both with and without animal contact [67,68]. The unregulated use of antimicrobials in animal and human medicine may result in increased drug resistance posing a threat to public health. Increased surveillance is required to understand the epidemiology of AMR bacteria in this region to develop appropriate integrated control strategies.”

ii)                Tables and figures should be described to be stand-alone (figure/table legends)

Details have been added to the tables and figures

iii)              Staphylococcus aureus should be italic throughout the manuscripts (See the attached file)

This has been corrected

iv)              Also all genes should be in italics (See the attached file).

This has been corrected

v)                All references should be revised according to the journal style.

This has been revised

Other minor comments, please see the attached file

These have been corrected -S. aureus and gene names have been italicised and we have added details about animal sources of S. aureus to the discussion

Reviewer 2 Report

The manuscript “Characteristics of Staphylococcus aureus isolated from patients in Busia County Referral Hospital, Kenya” by Obanda et al. evaluated the frequency of MSSA and MRSA nasal colonization in hospital patients from divers wards, their antibiotic resistance, virulence genes and genetic population structure. Considering that staphylococcal nasal colonization may lead to fatal consequences in a hospital environment, knowledge of their epidemiology is crucial. Studies undertaken by the Authors are valuable and interesting. The manuscript is well written, and the relevant data was presented in a clear, concise manner. The title reflects the purpose and results of the paper. The summary has all the necessary components, the methodologies used were referenced and the quality controls were cited in the text. However, some data need to be clarified or added to the manuscript.

Concerns:

  1. S. aureus nasal carriage rate identified by the Authors (11%) was relatively low, while basically, it is at least 30%. The reasons for this should be further explored in the Discussion. In the reviewer's opinion, this may have been due to culture of nasal swabs only on selective Mannitol Salt Agar and not on enriched media, such as Columbia agar.
  2. The clinically relevant and valuable strand of the manuscript concerning the empirical antibiotic therapy used in patients colonized with S. aureus was unfortunately not linked in any way to the antibiotic resistance data of the isolated S. aureus strains. If this cannot be completed in the Results, it should be raised at least in the Discussion.
  3. Some typos should be corrected, gene names should be in italics everywhere (e.g. sasX gene, lines 35 and 53). 

Author Response

Many thanks for your comments. We have responded below and in the manuscript

  1. S. aureus nasal carriage rate identified by the Authors (11%) was relatively low, while basically, it is at least 30%. The reasons for this should be further explored in the Discussion. In the reviewer's opinion, this may have been due to culture of nasal swabs only on selective Mannitol Salt Agar and not on enriched media, such as Columbia agar.

Details have been added to the discussion about the limitations of mannitol salt agar

2. The clinically relevant and valuable strand of the manuscript concerning the empirical antibiotic therapy used in patients colonized with S. aureus was unfortunately not linked in any way to the antibiotic resistance data of the isolated S. aureus strains. If this cannot be completed in the Results, it should be raised at least in the Discussion.

Details have been added to the results comparing MDR in patients who received antibiotics and those who did not receive antibiotics. We have also added details to the discussion

3. Some typos should be corrected, gene names should be in italics everywhere (e.g. sasX gene, lines 35 and 53). 

These have been corrected